5

## A global high-resolution map of debris on glaciers derived from multitemporal ASTER images

Orie Sasaki<sup>1</sup>, Omi Noguchi<sup>2</sup>, Yong Zhang<sup>2</sup>, Yukiko Hirabayashi<sup>2</sup>, Shinjiro Kanae<sup>1</sup>

<sup>1</sup>Department of Civil and Environmental Engineering, Tokyo Institute of Technology, 2-12-2 Ookayama, Meguro-ku, Tokyo, 153-8550, Japan

<sup>2</sup>Institute of Engineering Innovation, The University of Tokyo, 2-11-16 Yayoi, Bunkyo-ku, Tokyo, 113-8656, Japan

Correspondence to: Y. Hirabayashi (hyukiko@sogo.t.u-tokyo.ac.jp)

Abstract. Supraglacial debris affects the response of glaciers to climate change by altering the reflectivity of solar radiation and conductive heat flux. To accurately assess the contribution of glacier melts to sea level rise, water resources and natural hazards, it is important to account for the effects of debris. However, due to the practical difficulties of global-scale field measurements, information regarding the spatial distribution of the thickness and thermal properties of debris on glaciers is limited; hence, the effects of debris on glacier melting are not explicitly taken into account in current global glacier models. In this study, we developed a dataset of the thermal resistance of debris on glaciers at 90-m resolution derived from multi-temporal

- satellite images and satellite-derived radiation data at the global scale, excluding Greenland, Antarctica, and some of the Arctic. We found that supraglacial debris covered 16.8% of the entire analyzed glacial area. The highest debris cover percentage occurred in New Zealand, and the lowest was in Iceland. The area of thick debris (which suppresses glacier melting) was about two times that of thin debris (which accelerates glacier melting), indicating that the insulation effect of debris to inhibit glacier melting may dominate at the global scale. The distribution of debris was also related to the slope aspect of glaciers. Despite
- the limitations of this study, the resulting global distribution of the thermal resistance of debris can be incorporated into global glacier models, and hence it provides a solid basis for evaluating the effects of debris on glacial melting.

#### **1** Introduction

The ongoing retreat of small glaciers and ice caps around the world has many important impacts on human society (IPCC, 2013). Melting from small glaciers and ice caps separate from the two ice sheets in Greenland and Antarctica is a major

- contributor to sea level rise, despite the fact that they store less than 1% of global land ice (Kaser et al., 2006; Cogley 2009a,b; Gardner et al., 2013). These glaciers act as important regulators of regional water resources, especially in arid and semi-arid regions (IPCC, 2013; Immerzeel et al., 2010, 2012; Kaser et al., 2010). Meltwater from glaciers often leads to natural hazards such as rockslides and glacial lake outburst floods at regional scales (Richardson and Reynolds, 2000; Bolch et al., 2008a; Fujita et al., 2013). Hence, there is an increasing need to estimate the mass loss of individual glaciers worldwide and assess
- future sea level rise, water resources, and natural hazards, at both global and regional scales. To estimate current and future

changes in glaciers, several global glacier models have been developed in the past decade (Raper and Braithwaite, 2006; Hirabayashi et al., 2010, 2013; Radic and Hock, 2011; Marzeion et al., 2012; Slangen et al., 2012; Giesen and Oerlemans. 2013; Radic et al., 2014; Huss and Hock. 2015). Due to the high requirement of input data and computational costs, complex mechanisms of glacier retreat, such as the effects of debris on melting rate were not explicitly taken into account in these models.

Supraglacial debris is widely spread in many high-relief mountains, and has a potentially substantial influence on glacier behavior. The debris can accelerate (if debris is thin) or suppresses (if debris is thick) melt beneath the debris relative to clean ice (e.g., Østrem 1959; Mattson et al., 1993). Because debris-covered glaciers generally contain large ice volumes (Paul et al., 2004; Scherler et al., 2011; Benn et al., 2012), adequate modeling of debris-covered glaciers is important to realistically predict

future water availability and sea-level change (Bolch et al., 2008b; Benn et al., 2012; Zhang et al., 2013). Data regarding the distribution of debris that can be incorporated into glacier models is therefore key for improving the accuracy of current global glacier models.

Glacial melting beneath debris is controlled by the spatial distribution of the debris thickness and thermal conductivity of the debris, which can be obtained from ground observations. Several existing glacier mass balance models have successfully

- represented ice melting under debris at the local scale by incorporating *in situ* measurements of the debris layer (e.g., Nicholson and Benn, 2006; Reid and Brock, 2010; Reid et al., 2012; Rounce et al, 2015). However, these models were only applied to individual glaciers at local scales because it is unrealistic to measure such parameters over a large area. Although studies using satellite data have presented supraglacial debris mapping on a relatively large scale (~40 km), they have focused only on the spatial distribution of debris cover, or debris thickness in limited specific areas, and have not estimated the effects of debris on
- glacier melting because of the lack of data on the thermal conductivity of the debris layer (e.g., Lambrecht et al., 2011; Mayer et al., 2011; Casey et al., 2012; Frey et al., 2012). Hence, the creation of a global-scale debris map including thermal variables remains to be achieved.

In this study, we developed a global dataset of the spatial distribution of debris on glaciers, as well as information on thermal variables from satellite data. As mentioned above, it is difficult to determine debris thickness and the thermal conductivity of

- the debris layer without field measurements. Therefore, we focused on the 'thermal resistance', defined as the debris thickness divided by the thermal conductivity of the debris layer. The concept of thermal resistance, proposed by Nakawo and Young (1981) was convenient to obtain the thermal variables of debris on glaciers only from satellite data. Nakawo and Rana (1999) confirmed that thermal conductivity obtained from Landsat satellite data reasonably replicated the effect of debris on glacial melting in the Lirung Glacier in the Nepalese Himalayas. The thermal resistance can be used to estimate the effects of debris
- on the ablation rate of glaciers because it is a key parameter used to calculate the insulation energy from the atmosphere to ice. Zhang et al. (2011) demonstrated that satellite-derived thermal resistance values accurately reflected large-scale variations in the extent and thickness of the debris cover in *in situ* measurements at the Hailugou Glacier on the Tibetan Plateau. Furthermore, because thermal resistance is temporally consistent in different seasons and years (Suzuki et al., 2007), it is useful as a parameter in glacial melt models.

of glacier evolution that includes debris effects.

Building on previous studies that estimated thermal resistance from satellite images in several limited-size local glaciated areas (Rana et al., 1997; Nakawo and Rana, 1999; Suzuki et al., 2007; Zhang et al., 2011; Fujita and Sakai, 2014), and using newly available global glacier outlines and multiple satellite images, we determined the spatial distribution of the thermal resistance of supraglacial debris at high resolution (90 m) on a global scale (with the exception of glaciers in Greenland, Antarctica, and some parts of the Arctic). The data we obtained provides key information for developing global-scale modeling

5

## 2 Data and Method

#### 2.1 Data

To determine the boundaries of each glacier, we used the Randolph Glacier Inventory (RGI), version 4.0 (Arendt et al., 2014).
This is a globally complete inventory of glacier outlines (Arendt et al., 2014) that provides locations (longitude and latitude), maximum and minimum altitudes, and glacier areas. The inventory is divided into 19 regions, 13 of which were considered in this study (Fig. 1). We excluded the remaining six regions (Arctic Canada (North), Arctic Canada (South), Greenland, Russian Arctic, North Asia, and Antarctica) due to interference from snow and clouds in the satellite data.

- A Level 3A01 product of Advanced Spaceborne Thermal Emission and Reflection Radiometer (ASTER) orthorectified images and downward radiation products of the Fast Longwave and Shortwave Fluxes (FLASHFlux) project were used to determine the spatial distribution of the thermal resistance of debris on glaciers. ASTER measures three visible and nearinfrared (VNIR) bands (0.5–0.9 µm) at 15-m spatial resolution, six short-wave infrared (SWIR) bands (1.6–2.4 µm) at 30-m spatial resolution, and five thermal infrared (TIR) bands (8.1–12 µm) at 90-m spatial resolution. We used the ASTER VNIR and TIR bands, and selected 4,264 images acquired between January 2009 and December 2013, using the criterion that cloud
- 20 cover in each image was less than 40%. The selected images covered 85.8% of the glaciers in the 13 RGI target regions. The region 'Low Latitudes' covered 71.9%, which was the lowest percentage, while the region 'New Zealand' covered 99.8%, which was the highest (Table 1). In addition, we used ASTER Digital Elevation Model (DEM) data to investigate the characteristics of debris distribution. The ASTER DEM is generated from VNIR nadir-looking (3N, 0.76–0.86 µm) and background-looking (3B, 0.76–0.86 µm) telescopes.
- 25 FLASHFlux is a product of the Clouds and the Earth's Radiant Energy System (CERES) project, which releases top-ofatmosphere and surface radiative flux data at 1-degree resolution with a 1-week lag time from the observation date (Wielicki et al. 1996; Kratz et al. 2014). The CERES FLASHFlux data for downward shortwave and longwave radiation fluxes at the times closest to ASTER data acquisition were obtained between January 2009 and December 2013. Details of the CERES FLASHFlux data can be found in Kratz et al. (2014) or at http://flashflux.larc.nasa.gov/.

#### 2.2 Thermal resistance of debris layers

Thermal resistance was obtained from ASTER images, FLASHFlux radiation data, and RGI. Figure 2 shows a schematic diagram of the calculation. The thermal resistance of a debris layer was defined as the debris thickness divided by the thermal conductivity of the debris layer (Nakawo and Young, 1981, 1982), as follows:

5 
$$R = \frac{h}{\lambda}$$

30

(1)

where *R* is the thermal resistance (K m<sup>2</sup> W<sup>-1</sup>), *h* is the debris thickness (m), and  $\lambda$  is the thermal conductivity of the debris layer (W m<sup>-1</sup> K<sup>-1</sup>).

We calculated the thermal resistance of the debris layer based on the energy balance at the debris surface, which was a similar approach to that used in previous studies (e.g., Suzuki et al. 2007; Zhang et al. 2011; Zhang et al. 2013; Fujita and

Sakai 2014). Following these studies, we simplified the calculations by (1) assuming a linear temperature gradient within the debris layer, (2) allowing for no heat storage in the debris layer, and (3) using a conductive heat flux ( $Q_G$ ; W m<sup>-2</sup>) as the heat flux reaching the ice beneath the debris. From the first assumption,  $Q_G$  can be calculated from the debris surface temperature ( $T_S$ ; °C), debris–ice interface temperature ( $T_i$ , assumed to be the melting point of water, 0°C), and thermal resistance (R) of the debris layer, as follows:

$$15 \quad Q_G = \frac{T_S - T_i}{R} \tag{2}$$

From the second and third assumptions,  $Q_G$  can be described as a residual term of the energy balance at the debris surface, as follows:

$$Q_G = S \downarrow (1 - \alpha) + L \downarrow -\varepsilon \sigma T_S^4 + LE + H$$
(3)

where  $S \downarrow$  is the downward shortwave radiation (W m<sup>-2</sup>),  $\alpha$  is the albedo of the debris surface, L  $\downarrow$  is the downward longwave

- 20 radiation (W m<sup>-2</sup>), ε is the emissivity (taken to be 1), σ is the Stefan-Boltzmann constant ( $5.67 \times 10^{-8}$  W m<sup>-2</sup> K<sup>-4</sup>), *T<sub>s</sub>* is the surface temperature (°C), *LE* is the latent heat flux (W m<sup>-2</sup>), and *H* is the sensible heat flux (W m<sup>-2</sup>). Previous studies found that, especially with clear sky conditions, the contribution of turbulent heat fluxes to the total energy balance was typically small on debris-covered glaciers (Mattson and Gardner 1989; Takeuchi et al., 2000; Suzuki et al., 2007; Zhang et al., 2011). Due to this, the turbulent heat fluxes (*LE*,*H*) were assumed to be zero because we only used images with clear sky conditions.
- From Eq. (1) and (3), the thermal resistance R was calculated as follows:

$$R = \frac{T_S}{S \downarrow (1-\alpha) + L \downarrow -\varepsilon \sigma (T_S + 273.15)^4}$$
(4)

The brightness temperatures on the glaciers were retrieved from five ASTER TIR bands, and were used as the surface temperatures,  $T_S$  (Alley and Nilsen, 2001). The albedo was directly calculated from the spectral reflectance at the top of the atmosphere in three VNIR ASTER bands (Yüksel et al., 2008). The spatial resolution of the thermal resistance was constrained by the resolution (90 m) of the ASTER TIR sensors.

We obtained the thermal resistance of all available images using the above method

We obtained the thermal resistance of all available images using the above method and then selected the highest thermal resistance value in each grid cell where multi-temporal images were available to calculate thermal resistance. This process

excluded pixels with clouds and snow cover because thermal resistance would be underestimated due to interference. If the calculated thermal resistance was unrealistically high (>  $30 \text{ m}^2\text{KW}^{-1}$ ), the pixel was excluded prior to the selection of the highest value. This threshold was subjectively decided by comparing the calculated thermal resistance and visible images of glaciers from several regions. Finally, the resulting thermal resistance values were overlaid onto the RGI 4.0 glacier outline data and a global-scale distribution map of thermal resistance on the glaciers with 90-m resolution was created. These processes

5

are schematically summarized in Fig. 2.

#### **3** Result

#### 3.1 Distribution of the thermal resistance of debris

We generated a global 90-m resolution map of thermal resistance estimates on glaciers. The total debris-covered area in the area of glaciers analyzed was 260,432 km<sup>2</sup>. Figure 3 shows examples of the distribution of estimated thermal resistance in two regions (Baltro Glacier in Pakistan and Southern Alps in New Zealand). For the same thermal conductivity, high and low thermal resistance indicated thick and thin debris, respectively. A thermal conductivity of zero indicated clean ice. A thick debris layer exists mainly at the terminus of glaciers and the layer becomes thinner with increasing elevation along the glacier. The percentages of debris-covered area to total glacier area were calculated to investigate the regional differences in debris

- distribution. We set the threshold of thermal resistance at 0.01 m<sup>2</sup> K W<sup>-1</sup> to differentiate between clean ice and debris-covered ice. This threshold is equal to a debris thickness of 3–10 mm, depending on the thermal conductivity. The percentage of debris within each region varied from 4.7 to 47.4%, with an average value of 16.8% (Table 2). Iceland and Svalbard had the lowest percentages of debris at 4.7 and 6.6%, respectively. New Zealand and the Caucasus and Middle East had the highest percentages at 47.4 and 34.8%, respectively. Relatively high percentages (21.3 and 26.1%) were observed in Central Europe
- and Central Asia, and a relatively low percentage (12.2%) was observed in the Southern Andes. In the remaining regions [Alaska, Western Canada and USA, Scandinavia, South Asia (West) and South Asia (East)], the percentage of debris-covered area varied from 15.1 to 18.6%, which were similar values to the global mean.

#### **3.2 Validation**

We validated the distribution of thermal resistance estimates by comparing our values with those of previous studies in various regions. Because a few studies estimated the distribution of debris thermal resistance, we also compared our results with the distribution of debris thickness and the extent of debris cover reported in previous studies.

#### 3.2.1 Distribution of thermal resistance

We first visually compared the derived thermal resistance values with those of previous studies at selected glaciers, and examined whether the distribution of thermal resistance was consistent. If there was any discrepancy, we referred to additional

30 images such as Landsat and Google satellite images to examine the cause of the discrepancy. The distribution of the thermal

5

resistance of debris corresponded well with those of three previous studies that used an almost identical methodology of this study, but different satellite data, in the Lunana region (Bhutan; Suzuki et al., 2007), and at the Hailuogou Glacier (Tibetan Plateau; Zhang et al., 2011) and Trambau Glacier (Nepalese Himalayas; Fujita and Sakai, 2014). Our results from the Baltro Glacier (Karakoram) (Fig. 3a) were in close agreement with those of Mihalcea et al. (2008), who estimated thermal resistance by combining ASTER satellite images, meteorological data, and field measurements.

3.2.2 Distribution of debris thickness

Due to the limited number of studies estimating debris thermal resistance, our study and the spatial distribution of debris thickness from prior studies conducted at the Mirage Glacier (Italy; Foster et al., 2012), Franz Josef Glacier (New Zealand; Brock et al., 2013), Khumbu Glacier and others (Nepalese Himalayas; Rounce and McKinney, 2014), Koxkar Glacier
(northeastern China; Juen et al., 2014), and Bara Shigri Glacier (Indian Himalaya; Schauwecker et al., 2015) were qualitatively compared. For the comparison, we assumed uniform thermal conductivity within a glacier that produced a linear relation between thermal resistance and debris thickness. For all glaciers in this study, except for the Koxkar Glacier, where our methodology did not detect any debris, the general distribution of thick and thin debris in the previous studies corresponded closely with high and low values of thermal resistance, respectively. The potential reason of the discrepancy in the Koxkar
Glacier will be discussed in Section 4.3 below. In addition, our methodology was unable to detect small debris, such as center

Glacier will be discussed in Section 4.3 below. In addition, our methodology was unable to detect small debris, such as cente moraine in the Franz Josef Glacier, primarily due to limitations in the spatial resolution (90 m) of the satellite data.

#### 3.2.3 Extent of debris cover

We compared estimates of the extent of debris cover from this study and five previous studies based on field measurements or satellite images and using different methods to ours; Oberaletsch Glacier (Swiss Alps; Paul et al., 2004), Haut Glacier d'Arolla

- (Swiss Alps; Reid et al., 2012), Baltoro Glacier (Karakoram, Himalayas; Veetil et al., 2012), and glaciers in the northern Caucasus (Lambrecht et al., 2011), and central parts of the Southern Alps in New Zealand (Anderson et al., 2012). We selected thresholds of thermal resistance using the minimum values of the thickness and thermal conductivities from these studies, and of locations classified as debris-covered ice and clean ice to estimate the debris cover extent. For example, Reid et al. (2012) measured debris thicker than 1.0 cm and the calculated thermal conductivity was 0.94 Wm<sup>-1</sup>K<sup>-1</sup>. In this case, the threshold of
- thermal resistance was calculated as 0.011 m<sup>2</sup>KW<sup>-1</sup> from Equation (1). If no information on debris thickness or thermal conductivity was given, we subjectively set the threshold as 0.015 m<sup>2</sup>KW<sup>-1</sup>, corresponding to a thickness of ~0.5–1.5 cm, depending on the thermal conductivity value. Visual comparison of the extent of debris cover showed similar distributions at all five sites. Table 3 shows the percentage of debris-covered area to the entire glacier area, excluding the Baltoro Glacier, for which information was unavailable. The percentages of debris-covered areas were very similar in the Oberaletsch, Tasman,
- and Burton Glaciers, while our study overestimated values in the Haut Glacier d'Arolla and six glaciers in the Caucasus. In the Haut Glacier d'Arolla in particular, our estimated values were more than two-fold higher than the values in Reid et al.

(2012) obtained from a field survey. This discrepancy was due to the large debris-covered area of the southeastern tributary of the glacier (marked as a circle in Fig. 5a and b), which may not have been measured by the survey of Reid et al. (2012). The Landsat image of the southeastern tributary displayed a distribution of dark-colored ice, indicating that there may have been debris on the glacier, although Reid et al. (2012) reported no debris.

- Figure 7 compares the area-elevation distributions of debris-covered ice and clean ice from this study and Lambrecht et al. (2011) at six glaciers (see Fig. 6 for distribution of thermal resistance and locations; Shkhelda, Bzhedukh, Kashkatash, Bashkara, Djantugan, and Djankuat Glaciers) located in the Adyl-su Valley on the northern slope of the Caucasus Mountains. The clean ice and debris-covered ice were in good agreement at elevations less than 3000 m. However, they did not match at higher elevations. Debris-covered ice was present between 2300 and 4300 m in our results, whereas the estimates from
- Lambrecht et al. (2011) were limited to elevations between 2350 and 3200 m. These marked differences may come from the interference of snow on the glaciers. If snow cover affected the estimates, then supraglacial debris at high elevations would have been covered by snow and mislabeled as clean ice. Due to this factor, and most notably at high elevations, the extent of debris-covered ice in this study was greater than that of Lambrecht et al. (2011). The effect of snow cover could also affect the delineation of the outlines of glaciers from satellite data. The glacier outlines in Lambrecht et al. (2011) were more extended
- than those in our study obtained from RGI at high elevations, and the resulting glacier areas were larger than that of our study in the southwest area of the Djankuat Glacier and southeast area of the Shkhelda Glacier. One possible explanation for this may be that Lambrecht et al. (2011) used a single satellite image obtained in 2000 to estimate the glacier outline and distribution of debris on glaciers, whereas we used multi-temporal images to eliminate the potential interference of snow cover.

#### **4** Discussion

#### 20 4.1 Regional differences in the distribution of debris thickness

The effect of debris on glacial melting depend on the thickness of the debris layer. A thin debris layer accelerates ablation by reducing albedo, while a thick debris layer inhibits melt because energy conduction to the ice is reduced beneath the debris (e.g., Østrem et al., 1959; Driedger et al., 1981; Mattson et al., 1993). We therefore classified debris as thin (0.01–0.015 m<sup>2</sup>KW<sup>-1</sup>), intermediate (0.015–0.025 m<sup>2</sup>KW<sup>-1</sup>), and thick ( $\geq 0.025$  m<sup>2</sup>KW<sup>-1</sup>), and compared regional differences in the distribution of

- debris thickness (Table 2). These thresholds were determined from the mean and standard deviation of thermal resistance at a critical thickness, i.e., the debris thickness when the melting rate beneath debris equals that of clean ice, which was obtained from the literature (Table 4). The global average of thin, intermediate, and thick debris-covered areas to total glacier area were 3.6, 5.2, and 8.0%, respectively (Table 2 and Fig. 8). This indicated that, globally, the area of thick debris cover was almost double that of thin debris cover. The percentages of thick debris exceeded those of thin debris except for Svalbard and
- 30 Scandinavia. In particular, thick debris covered more than 50% of the total debris-covered area in the Caucasus and Middle East and in New Zealand.

#### 4.2 Geographical characteristics of debris distribution

Because the formation conditions of debris covered glaciers are affected by topographic variables such as aspect (Benn, 1989; Nagai et al., 2013) and steepness (Scherler et al., 2011), we examined the relationships between debris thickness and slope aspect and steepness in each RGI region. Because the debris-covered area was small, and it was difficult to investigate its geographical distribution, we excluded Iceland and Svalbard from the analysis. There was no clear relationship between steepness and distribution of debris cover (not shown), while a well-defined dependency of debris cover on aspect was found (Fig. 9). In the northern hemisphere, the debris-covered area was largest on south-facing or southeast-facing slopes, and in the southern hemisphere it was largest on north-facing or northeast-facing slopes. For example, in South Asia (East), the debris-covered ratio on south-facing slopes was almost three times that on north-facing slopes. The percentage of thin debris-covered

area was almost constant for all aspects, while the thick debris-covered area varied depending on aspect.

#### 4.3 Uncertainties

The main source of uncertainty in the thermal resistance of debris determined in this study was the quality of satellite images. Because Equation (3) could be applied only in clear sky conditions (Suzuki et al.,2007; Zhang et al., 2011), our method underestimated thermal resistance when snow or clouds were present. In addition, our methodology was able to detect debris

- thermal resistance only when ice under debris was at the melting condition [assumption of  $T_I = 0$ °C in Equation (2)]. Our failure to detect debris cover in the Koxkar Glacier demonstrated these limitations. Only two ASTER images (2009/11/2 and 2013/5/5) of the glacier could be used because most images did not pass the criterion of

Thermal resistance was more sensitive to downward shortwave than longwave radiation in most regions. Notably, Svalbard displayed by far the largest uncertainties, which may be due to the small debris-covered area (5.6%) and low values of thermal resistance in most debris in that location.

The detection of small debris such as center moraines was also limited due to the spatial resolution (90 m) of ASTER. In addition, inconsistencies in the glacier outlines between ASTER and RGI data due to the time differences caused additional errors in debris distribution estimates, particularly with rapidly melting glaciers. When ASTER images were more recent than RGI data, and if glaciers in ASTER had melted in recent years and showed bedrock within the RGI glacier outlines created from old data, debris was defined as thick in this study.

#### **5** Conclusion

- We created a global distribution map of thermal resistances of debris on glaciers (except in Antarctica and Greenland) derived from multi-temporal ASTER images obtained between 2009 and 2013 at 90-m resolution. Our results revealed that 16.8% of the total analyzed glacier area was covered by debris. The highest percentage of debris cover was found in New Zealand, and the lowest was in Iceland. In most regions, the areas of thick debris cover were larger than those of thin debris cover, with the exception of Svalbard and Scandinavia, indicating that the insulation effect of debris to inhibit glacier melting was dominant
- in debris-covered regions. We also found that the distribution of debris-covered areas was related to the of slope aspect of glaciers. Despite the limitations of this study, the resulting global distribution of the thermal resistance of debris on glaciers can be incorporated into global glacier models and, thus provides a solid basis for evaluating the effects of debris on glacial melting.

#### Acknowledgements

This paper was financially supported by the Funding Program for the Global Environmental Research Fund (S-10 and S-14) by the Ministry of the Environment, Japan and MEXT KAKENHI Grant Number 15K06228 and 16H06291.

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
