# Peer review of "A global high-resolution map of debris on glaciers derived from multitemporal ASTER images"

_The Cryosphere, 2016_

## Short Comment (SC1) · 2 Dec 2016

The debris thickness mapping effort on a global scale is a commendable and important area of research that will help understand the debris' influence on glacier melt in response future climate projections. There are many interesting aspects of this paper and one important contribution is determining the areal extent of debris cover throughout the world. This in itself is important. However, I believe the thermal resistance estimates need to be treated with tremendous caution as they appear to be orders of magnitude smaller than those that one would expect. This severe underestimation is likely due to the methodology used in this study, which will be discussed in more detail below.

Regarding the validity of the thermal resistance estimations, this study uses thermal resistances from Suzuki et al. (2007), Mihalcea et al. (2008), Zhang et al. (2011), and Fujita and Sakai (2014). Figure 3 shows a thermal resistance map of Baltoro Glacier, which the authors state was in close agreement with Mihalcea et al. (2008). However, if one assumes a thermal conductivity of 1 W m-1 K-1 (a reasonable value for the area), the terminus of Baltoro Glacier has a thermal resistance of $\sim$0.05 m2 K W-1, which is a debris thickness of $\sim$ 0.05 m. This is 1-2 orders of magnitude smaller than the debris thickness maps derived by Mihalcea et al. (2008) at the terminus, which ranged from 0.5 − 3 m. It is important to note that while these thermal resistances are severely underestimated, the trends in the spatial distribution of the thermal resistance (higher values toward the terminus that thin upglacier) agree well. This agreement of spatial trends, despite severely underestimating the actual values of thermal resistance, was noted by Rounce and McKinney (2014). This is likely a product of the model simply reflecting trends related to the surface temperature over the debris as one would expect.

The good agreement between this study and the thermal resistances derived from Suzuki et al. (2007), Zhang et al. (2011), and Fujita and Sakai (2014) is likely due to these models using the same methods and assumptions; however, those studies also appear to severely underestimate the thermal resistance as well. Suzuki et al. (2007) show thermal resistances for the Everest region (Figure 6 in that study) that have maximum values of $\sim$ 0.03 m2 K W-1 (or a maximum debris thickness of 0.03 m), which severely underestimate debris thickness measurements in the Everest region from Ngozumpa Glacier (Nicholson and Benn, 2012) and Imja-Lhotse Shar Glacier (Rounce and McKinney, 2014) that commonly exceed 0.30 m and even exceed 1 m as well. As previously mentioned, Zhang et al. (2011) and Fujita and Sakai (2014) make the same assumptions as Suzuki et al. (2007), which is why they also derived thermal resistances on the order of 0.03 m2 K W-1 (approximate debris thickness of 0.03 m), which are orders of magnitudes lower than those measured in the field in the Himalaya.

These thermal resistances are likely severely underestimated due to the assumptions

made in the methods. The first major assumption that influences these results is assuming that the sensible heat flux is negligible. On the contrary, the sensible heat fluxes are quite significant and can range from 100-400 W m-2 (e.g. Reid and Brock, 2010; Rounce et al., 2015). If one examines Equation 4, including the sensible heat flux would reduce the denominator, which would in turn increase the derived thermal resistance. This increase is often quite substantial. The second major assumption is that the temperature profile in the debris is linear. Debris temperature measurements have shown that at the time that the satellite images are acquired (early morning in the Himalaya), the temperature profiles are actually highly nonlinear, which can cause the thermal resistance to be underestimated by a factor of 2-3 (Rounce and McKinney, 2014). These two major assumptions both cause the thermal resistance to be severely underestimated.

I do want to express my support of this work as I believe mapping the thermal resistance (or debris thickness) of all glaciers would be a valuable contribution to the current state of knowledge; however, the severity of these underestimations is alarming and should be addressed.

References: Fujita, K. and Sakai, A.: Modelling runoff from a Himalayan debris-covered glacier, Hydrol. Earth. Syst. Sci., 18:2679-2694, 2014. Mihalcea, C., Mayer, C., Diolaiuti, G., D'Agata, C., Smiraglia, C., Lambrecht, A., Vuillermoz, E., Tartari, G.: Spatial distribution of debris thickness and melting from remote-sensing and meteorological data, at debris-covered Baltoro glacier, Karakoram, Pakistan, Annals of Glaciology, 48:49-57, 2008. Nicholson, L. and Benn, D.I.: Properties of Natural Supraglacial Debris in Relation to Modelling Sub-Debris Ice Ablation, Earth Surface Processes and Landforms, 38(5):490-501, 2012. Reid, T.D. and Brock, B.W.: An Energy-Balance Model for Debris-Covered Glaciers Including Heat Conduction through the Debris Layer, J. Glaciol., 56(199):903-916, 2010. Rounce, D.R. and McKinney, D.C.: Debris thickness of glaciers in the Everest area (Nepal Himalaya) derived from satellite imagery using a nonlinear energy balance model, The Cryosphere, 8:1317-1329. Rounce, D.R.,

Quincey, D.J., and McKinney, D.C.: Debris-covered glacier energy balance model for Imja-Lhotse Shar Glacier in the Everest region of Nepal, The Cryosphere, 9:2295-2310, 2015. Suzuki, R., Fujita, K., and Ageta, Y.: Spatial Distribution of Thermal Properties on Debris-Covered Glaciers in the Himalayas Derived from ASTER Data, Bulletin of Glaciological Research, 24:13-22, 2007. Zhang, Y., Fujita, K., Liu, S., Liu, Q., and Nuimura, T.: Distribution of Debris Thickness and its Effect on Ice Melt at Hailuogou Glacier, Southeastern Tibetan Plateau, Using In Situ Surveys and ASTER Imagery, J. Glaciol., 57(206):1147-1157, 2011.

---

## Referee Comment (RC1) · Anonymous Referee #1 · 19 Dec 2016

General Comments

The study of Sasaki et al utilises multi-temporal satellite thermal imagery from the ASTER sensor and radiation data from the CERES project, combined with glacier outlines in the Randolph Glacier Inventory, to estimate: (1) the global distribution (excluding polar regions) of supraglacial debris cover at 90 m resolution; and (2) the distributions of 'thick' (ablation inhibiting) and 'thin' (ablation enhancing) debris cover over the same areas. These are ambitious aims, but such a global dataset would be of enormous value to earth science, and studies of glacier response to climate change in particular. Unfortunately, the study contains basic flaws in the methodology, relating to mistaken and untested assumptions about the surface energy balance of debris

cover. These render the estimates of 'thick' and 'thin' debris cover distributions unreliable, while a much more rigorous evaluation of the critical threshold between clean ice and debris-covered ice would be needed to support the estimates of total debris covered area. These issues are explained in more detail below. Due to these significant problems, it would be misleading if the outputs from this study were incorporated into global glacier models to assess the effect of debris on glacier melt, as intended by the authors.

Specific Comments

Key Issue 1: Assumptions regarding the surface energy balance of supraglacial debris

Section 2.2 of the paper introduces several important assumptions which undermine the so called 'energy balance' approach in this study. Only one of these assumptions (that the debris-ice interface is at 0 degrees C.) is assessed later in the paper, although its implications are not fully realised in the discussion. The values of thermal resistance, R, which are the basis of the mapping methodology, are derived as an energy balance residual and in order to do achieve this, the energy balance needs to be as complete, consistent and as accurate as possible. The methodology fails to achieve this.

The assumptions of a) instantaneous linear temperature gradient, and b) no heat storage in the debris layer are known to be incorrect from several studies (e.g. Nicholson and Benn, 2006; Brock et al., 2010, Reid and Brock, 2010; Rounce and McKinney, 2014). These processes are spatially and temporally variable in magnitude, depending on debris thickness, thermal properties and weather conditions before and at the time of satellite imaging and yet these important assumptions are not revisited in the paper. This undermines the attempt to estimate areas of debris cover with different melt impacts (i.e. 'thick' and 'thin' debris) made in the paper.

The next assumption which is not evaluated is that the emissivity of debris = 1. Given rock forming minerals all have emissivity < 1 this assumption is clearly wrong, and even very small changes in emissivity of 0.01 have a significant impact on the calculated

longwave radiation flux estimated from the Stefan-Bolzmann relationship (equation 3). There is no acknowledgement of this problem or attempt to evaluate its impact on debris cover estimations in the paper.

Probably most critical is the assumption that turbulent fluxes = 0. The 4 references given here are selective, dated and do not provide support for this assumption. Recent, and more rigorous, field and modelling studies of the energy balance of supraglacial debris have been ignored (e.g. Nicholson and Benn, 2006; Brock et al., 2007, Brock et al., 2010, Reid et al., 2012; LeJenue et al., 2013; Collier et al., 2014; Fyffe et al., 2014; Rounce et al., 2015). These studies clearly demonstrate the opposite, i.e. that the sensible heat flux is a very large flux of energy away from the debris surface, particularly under the conditions when the imagery will have been captured, i.e. debris is being warmed strongly by insolation leading to a steep debris-air temperature gradient. Furthermore, the importance of including turbulent fluxes when mapping debris cover from thermal satellite imagery using an 'energy balance residual' approach has been demonstrated by several studies (Foster et al., 2012; Rounce and McKinnery, 2014; Schauwecker et al., 2015).

The assumption of zero turbulent fluxes creates an energy imbalance and a large underestimate of the thermal resistivity of debris covers (R; equation 4). This problem can be demonstrated by substituting the resulting R values (e.g. figures 3-5) into equation 1, using possible thermal conductivity (lambda) values of between 0.5 and 2, to estimate debris thickness. The resulting debris thickness is only in the range up to a maximum of 14 cm for lambda = 2, and 7 cm for a more realistic value of lambda = 1 (or 3.5 cm for lambda = 0.5). For example, on Baltoro glacier debris is estimated to be only a few cm over the majority of the glacier ablation zone (Figure 3) while it is known to exceed 100 cm on the lower tongue. The other examples in figures 3 to 6 show similarly unrealistically low debris thickness values if realistic thermal conductivity values are used.

Crucially, errors due these assumptions will be spatially and temporally variable due to

variable meteorological conditions (particularly wind speed and air temperature) and debris thickness and thermal properties. This renders not only the resulting R values physically meaningless and unusable, but also makes relative comparisons between different glaciers, and even different sites on the same glacier, impossible. Hence, the key aim of the paper, to produce a global map of the thermal resistance of debris for use in global glacier models is not achieved. The 'headline' claim that the global area of thick debris (ablation inhibiting) is two times the area of thin debris (ablation enhancing) is unsupported by the analysis.

Key Issue 2: Threshold thermal resistance between clean ice and debris-covered ice

This threshold R value is set to 0.01 m^2 K W^-1 (p. 5, line 15). Note that, given the problems discussed under Key Issue 1, there is large uncertainty in derived R values and this critical value will not provide a globally-consistent clean ice-debris cover threshold between different glaciers and regions. Given that this threshold value is critical to the estimates of total debris covered area, there is surprisingly little discussion of what this value means in terms of actual debris cover, and what the implications of uncertainty in the critical threshold are. What is the effect of changing the R threshold to 0.02, or to 0.001, for example? The value of 0.01 sounds like a convenient number, rather than one that is based on sound scientific reasoning. The upper ablation zones of debris covered glaciers typically have large areas of dirty ice, patchy debris, and thin debris cover which grades into mostly continuous debris cover down glacier. It is not clear what the 0.01 threshold corresponds to in this transition. This is an important issue as thin and patchy debris is likely to lead to increased melt rates compared with clean ice, whereas continuous debris will normally lead to a reduction in ablation. Uncertainty in where this cut-off lies undermines the estimates of total debris covered area, and further weakens the attempt to calculate relative areas of 'thick' and 'thin' debris.

Other issues

The evaluation of debris covered areas in figures 3-6 shows some general agreement between the present study and earlier work, but also important discrepancies, particularly missing medial moraines. The explanation for the 'missing' debris cover at Koxkar Glacier (some 18 kmˆ2 of debris and 60% of the ablation zone according to Juen et al., 2014) discussed in Section 4.3 is reasonable, but this does demonstrate the problem that, for high elevation continental glaciers, overnight freezing of debris is common (particularly under clear sky conditions) which is a significant issue for the energy balance method of debris thickness estimation as it leads to non-linear temperature profiles in debris and significant changes in energy stored debris (both sensible and latent) as the debris warms in the morning (when satellite data are acquired). These processes are unaccounted for in the methodology.

Technical corrections

Overall, the study is well presented and concisely written. Given the significant problems with this work discussed above, however, I have not included a list of technical corrections in this review.

References not cited in the manuscript

Brock BW, Mihalcea C, Kirkbride MP, Diolaiuti G, Cutler M and Smiraglia C: Meteorology and surface energy fluxes in the 2005-2007 ablation seasons at Miage debris-covered glacier, Mont Blanc Massif, Italian Alps. Journal of Geophysical Research, 115, D09106, doi:10.1029/2009JD013224, 2010.

Collier E, Nicholson LI, Brock BW, Maussion F, Essery R and Bush ABG: Representing moisture fluxes and phase changes in glacier debris cover using a reservoir approach. The Cryosphere, 8, doi: 10.5194/tc-8-1429-2014, 2014.

Fyffe CL, Reid TD, Brock BW, Kirkbride MP, Diolaiuti G, Smiraglia C and Diotri F: A distributed energy-balance melt model of an alpine debris-covered glacier. Journal of Glaciology, 60(221), 587-602, doi:10.3189/2014JoG13J148, 2014.
LeJeune Y, Bertrand, J-M, Wagnon, P and Morin S: A physically based model of the year-round surface energy and mass balance of debris-covered glaciers. Journal of Glaciology, 59(214), doi: 10.3189/2013JoG12J149, 2013.

---

## Referee Comment (RC2) · F. Paul (Referee) · 27 Dec 2016

**General comments**

The study by Sasaki et al. applies a method to classify debris-covered glaciers as developed in earlier studies to a large part of the global glacier inventory as available from the RGI (Randolph Glacier Inventory). In contrast to numerous other studies, the aim here is not to classify the debris-covered parts itself, but to distinguish between clean ice and debris-covered ice within given glacier boundaries. The resulting dataset is going beyond a simple debris yes/no map and instead provides distributed thermal resistance of the glacier-covered area. This information is most welcome for a large number of applications looking at the energy balance for calculations of mass balance and future glacier development. I am also fine with the simplified approach suggested here, as achieving near-complete global coverage within a reasonable computational effort must have drawbacks somewhere. For these reasons I find the study timely and an important contribution.

On the other hand, I find some major shortcomings in this study that are partly based on problems of already published earlier studies, including some that are not cited here. In part and indirectly, the other two reviewers have mentioned the problem as well. The main issue is the never clearly defined and increasingly misleading use of the term debris-covered glacier. The term is used widely and independent of the spatial completeness of the debris coverage or its thickness. To my knowledge neither a glacier with a medial moraine nor with some more debris near the terminus should be called a debris-covered glacier. This requires that larger parts (tbd) of the ablation area have a near-complete (tbd) coverage with optically thick pebbles, stones or rock. Millimetre-sized particles (sand, silt, clay, BC, pollutants) that are often creating the rough microstructure of the ice do not fall into this category, as they never cover the ice completely. A side issue of this problem is that the aggregation of often highly variable thermal information at the level of a 90 m thermal pixel is not addressed in most studies, e.g. by looking at least at the co-registered spectral information at 15 and 30 m resolution (speaking of ASTER). In consequence, an increasing amount of clean ice (towards higher elevations) within a 90 m pixel is often confused with apparently decreasing thickness of debris cover (although a glacier is never sorting particle sizes).

I exemplify the problem here for Hailuogou Glacier (HG), as it has been used for the methodological development that is forming the base of this study (Zhang et al. 2011). HG might be considered as a debris-covered glacier, but in fact the (optically thick) debris is only covering its lowermost part. For the largest part of the ablation area the heavily crevassed surface is *dirty* but not really debris covered. On a micro-scale, the dirty ice is not covering the ice completely and a variable amount of zero-degree bare ice is contributing to the thermal signal when aggregated at 90 m pixels. For HG this issue is strongly enhanced due to the highly crevassed surface that adds further zero-degree zones to the 90 m pixel. In fact, the 15 m ASTER VNIR image shown in Fig. 3a by Zhang et al. (2011) also clearly reveal (despite the resolution limits) that HG is not really debris covered but the ice is dirty at best (see also tourist photos in Google Earth). Accordingly, the thermal information leads to a very low thermal resistance and high melt rates (Fig. 6a/b). My key question here is: How could the thermal resistance derived from a dirty/crevassed glacier serve as a base for characterizing resistance for really debris-covered glaciers (let alone to determine debris thickness)?

When looking at Fig. 6 in Zhang et al. (2011) there is another severe issue in their analysis: A large portion (maybe not all) of the pixels indicating a high thermal resistance (the red-orange

band) are actually located outside the glacier boundary and cover the lateral moraine (and regions above) that are especially warm due to their south-easterly exposition and steep slope, i.e. the sun might hit these surfaces under a zero-degree incidence angle. There is definitely no ice underneath this "debris" that cools it. In consequence, the derived equations / regressions make no sense and could not be applied. This mistake highlights another problem with the (coarse) 90 m pixel of the thermal band. Not a single 90 m pixel with parts outside the glacier boundary should have been used to determine thermal information, as even a very small part of this warm rock impacts considerably on the mean value for the 90 m pixel (the same applies in the other direction when some bare ice is included). When the determination of debris thickness on a glacier is derived from rocks in the lateral moraine and an unconsidered bare ice part in the 90 m pixels, I do not wonder about funny results. At least a scientific base is missing.

I agree that this might not be the correct place to criticise the former study by Zhang et al. (2011) or note that the peer-review system fails from time to time. But when such studies are used as a base for other studies so that their misleading results are multiplied, there should be a possibility to stop it. I am aware that the above might have consequences for several other already published studies but I do not ask here to withdraw them. However, I would highly appreciate if all scientists working on the thermal identification of debris-covered glaciers (debris extent, thickness, reflectance or whatever) would do it more carefully in the future and consider all effects playing a role. This includes ice cliffs, melt ponds, albedo variations, shadow, crevasses, clean ice between thick debris (in regions of incomplete coverage) and the accurate distinction between dirty ice and really debris-covered ice. All these features can be present within the limits of a single 90 m ASTER pixel.

My other main objection is the rather poor validation performed here. I fully appreciate that several examples are given to illustrate the performance of the method (Figs. 3-6), including those where the method did not work. But the pure visual comparison gives a very unreliable base for a proper assessment. For example Baltoro Glacier in Fig. 3a: When I compare the thermal resistance map with a map of the pattern of clean ice / debris cover (nicely arranged in parallel medial moraines), I do not see any similarities. Upwards of the confluence area (Concordia) everything is completely blue despite several thick medial moraines, whereas rather clean ice (or again regions outside the glacier?) has a high thermal resistance. In the upper right it seems the glacier mask includes a larger rock outcrop (please note that the region in the red circle on Fig. 5 is also a rock outcrop; the glacier in this region melted away). To me it seems as if a satellite scene with a high amount of seasonal snow has been used for the classification, largely underestimating the real extent of debris cover. I am pretty sure this problem is prevalent also in many other regions.

Which brings me to a final major point, the selection of satellite scenes. I doubt that (a) the method applied here to select the most suitable scenes has always found the scenes with a minimum of snow cover and (b) I think for many regions suitable scenes simply do not exist (e.g. Fig. 6b). The selection of the scene with the largest amount of debris (P8, L21) is certainly a good idea but it does not imply that all debris is exposed. I think the uncertainty in this regard is not realistically estimated. I certainly miss a pixel-by-pixel comparison (omission and commission errors) based on a couple of manually created debris extents. This can be easily achieved by subtracting a clean ice classification (using a simple red/SWIR band ratio) from the RGI glacier extents (please use RGI 5.0!) converted to a grid. Such a clean ice mask would also help to determine the amount of clean ice within each 90 m thermal pixel and correct the details of the thermal resistance classification accordingly.

Overall, I think the study was worth a try as it ultimately creates a dataset of high interest and demand. Given that the methodology is further improved, better validated and creating more realistic results, I am pretty sure that the study can be published. I would thus like to encourage the authors to check if they can improve their study along the suggested lines and resubmit it at a later stage. In the following I add some further points requiring consideration in the case the authors intend to resubmit the study.

**Specific comments**

For future submissions, please use a continuous line numbering scheme and apply it to all lines rather than each 5th. This facilitates the work of reviewers greatly as they do not have to waste time with counting lines and page numbers.

P1

L17-19: As a more general comment: Please note that the use of the simplified terminology 'thin and thick debris' is misleading as it does not consider grain size and degree of coverage. The 'thin debris' that is enhancing surface melt is often just dirty ice (sand, clay, silt) with lots of clean ice in-between (i.e. incomplete spatial coverage). Moreover, for thin plates of rock (<2 cm) the albedo of the material is increasingly important. Please also note that surface melt is not the only factor contributing to volume/mass loss of glaciers and thus the amount of melt water. Several studies using DEM differencing have shown that specific mass loss of heavily debris-covered glaciers is often as high as for clean glaciers. Ice cliffs, melt ponds and so far completely unconsidered en-glacial melt (internal collapse of conduits) might play a major role for this. In short, the impact of debris cover on the amount of melt water from glaciers is not fully clear.

L24: Please just write 'glaciers'. Those contributing to sea level are certainly not 'small' but very large.

L28: How does melt water from glaciers cause rock slides?

P2

L4: Please note that glacier retreat (change in length) is something different than volume loss caused by the melting rate. Glaciers can lose mass without a change in length and advance without a change in mass. Global glacier models have a good hand on melt rates but difficulties with geometric changes, as this requires precise knowledge of the glacier bed (ice thickness distribution).

L7: As mentioned above, I would not call the material that is enhancing melt 'thin debris' but dirty ice, as grain size must be small, albedo low and its distribution disperse.

L13: Please use glacier instead of glacial when reference is made to contemporary glaciers (or here The melting of ice beneath debris …).

L31: Please see my comments above on the study by Zhang et al. (2011). I think it contains major systematic and methodological errors and cannot be used as a reference.

P3

L4: As the thermal resistance map is obviously not able to distinguish clean ice from dirty ice and debris-covered ice, what about starting with a simple debris cover yes-no map?

L9: Please use RGI 5.0.

P5

L14/15: The threshold values of thermal resistance for debris yes/no and thick/thin should be provided in the methods section.

P6

L4: Can 'close agreement' be described in more detail? I did not find a very good agreement between the thermal resistance map and the distribution of clean ice and debris for Balto-ro glacier (see General comments).

P7

L1: This is not a debris-covered tributary but bare rock. That this region has been mapped as debris-covered ice might be a consequence of the calibration over bare rock rather than debris-covered ice.

L11: Please note, debris cover should only appear on the surface below the ELA (or as a proxy: mean elevation) due to emergent flow. Apart from the higher amount of snow cover in the study by Lambrecht et al., I assume that your method has simply mapped dirty ice as being debris covered. Dirty ice is rather common in this region below steep (and ice free) rock walls.

L22: by reducing albedo? I do not think that albedo reduction is the major process here. The key point is that the material on the surface can get warmer than zero degrees and when the material is thin enough, conductivity can get the base of the material also above zero degrees so that it can melt into the ice.

L23ff: This sounds like methods rather than discussion.

P9

L5: Please note that also the RGI outlines are not perfect and please avoid including any thermal pixels that reach beyond given outlines. Only 100% inside pixels should be used for the calculation.

P17, Fig. 3: I suggest adding also here optical images for comparison (those with minimum snow amount).

---

## Author Comment (AC1) · 15 Mar 2017

We would like to thank the reviewers for their time to thoroughly review this submission. We have tried to respond to each comment and have incorporated valuable suggestions into an improved manuscript. Our one-by-one responses to comments are given in the attached document.

Please also note the supplement to this comment:
http://www.the-cryosphere-discuss.net/tc-2016-222/tc-2016-222-AC1-supplement.pdf
* * *

---

## Author Comment (AC2) · 15 Mar 2017

**Author's Response to Reviewer's Comments:**

**A global high-resolution map of debris on glaciers derived from multi-temporal ASTER images**

We would like to thank all reviewers for providing their constructive and very helpful comments. Here we address how we have revised the manuscript corresponding to reviewer's comments. The comments are written in blue color, which are followed by our replies written in black color.

**Response to Dr. David Rounce:**

(Short comment #1)

The debris thickness mapping effort on a global scale is a commendable and important area of research that will help understand the debris' influence on glacier melt in response future climate projections. There are many interesting aspects of this paper and one important contribution is determining the areal extent of debris cover through-out the world. This in itself is important. However, I believe the thermal resistance estimates need to be treated with tremendous caution as they appear to be orders of magnitude smaller than those that one would expect. This severe underestimation is likely due to the methodology used in this study, which will be discussed in more detail below.

Regarding the validity of the thermal resistance estimations, this study uses thermal resistances from Suzuki et al. (2007), Mihalcea et al. (2008), Zhang et al. (2011), and Fujita and Sakai (2014). Figure 3 shows a thermal resistance map of Baltoro Glacier, which the authors state was in close agreement with Mihalcea et al. (2008). However, if one assumes a thermal conductivity of 1 W m-1 K-1 (a reasonable value for the area), the terminus of Baltoro Glacier has a thermal resistance of ～0.05 m2 K W-1, which is a debris thickness of ～0.05 m. This is 1-2 orders of magnitude smaller than the debris thickness maps derived by Mihalcea et al. (2008) at the terminus, which ranged from 0.5 – 3 m. It is important to note that while these thermal resistances are severely underestimated, the trends in the spatial distribution of the thermal resistance (higher values toward the terminus that thin upglacier) agree well. This agreement of spatial trends, despite severely underestimating the actual values of thermal resistance, was noted by Rounce and McKinney (2014). This is likely a product of the model simply reflecting trends related to the surface temperature over the debris as one would expect.

The good agreement between this study and the thermal resistances derived from Suzuki et al. (2007), Zhang et al. (2011), and Fujita and Sakai (2014) is likely due to these models

using the same methods and assumptions; however, those studies also appear to severely underestimate the thermal resistance as well. Suzuki et al. (2007) show thermal resistances for the Everest region (Figure 6 in that study) that have maximum values of ～ 0.03 m2 K W-1 (or a maximum debris thickness of 0.03 m), which severely underestimate debris thickness measurements in the Everest region from Ngozumpa Glacier (Nicholson and Benn, 2012) and Imja-Lhotse Shar Glacier (Rounce and McKinney, 2014) that commonly exceed 0.30 m and even exceed 1 m as well. As previously mentioned, Zhang et al. (2011) and Fujita and Sakai (2014) make the same assumptions as Suzuki et al. (2007), which is why they also derived thermal resistances on the order of 0.03 m2 K W-1 (approximate debris thickness of 0.03 m), which are orders of magnitudes lower than those measured in the field in the Himalaya.

These thermal resistances are likely severely underestimated due to the assumptions made in the methods. The first major assumption that influences these results is assuming that the sensible heat flux is negligible. On the contrary, the sensible heat fluxes are quite significant and can range from 100-400 W m-2 (e.g. Reid and Brock, 2010; Rounce et al., 2015). If one examines Equation 4, including the sensible heat flux would reduce the denominator, which would in turn increase the derived thermal resistance. This increase is often quite substantial.

➢ Thank you for your comments. We have addressed all major comments one by one.
➢ We agree that sensible heat flux (and latent heat flux) should be considered. We therefore have updated our calculation incorporating them using global climate data, which is a bias-corrected reanalysis product of all required climate variables (temperature, relative humidity and wind velocity). As a result, for example, maximum value of thermal resistance in Baltoro glacier increases 44% (from 0.09 to 0.13). As you indicated, overall distribution of thermal resistance does not change in the new calculation, but underestimation of thermal resistance was improved in many regions.

The second major assumption is that the temperature profile in the debris is linear. Debris temperature measurements have shown that at the time that the satellite images are acquired (early morning in the Himalaya), the temperature profiles are actually highly nonlinear, which can cause the thermal resistance to be underestimated by a factor of 2-3 (Rounce and McKinney, 2014). These two major assumptions both cause the thermal resistance to be severely underestimated.

➢ Since the ASTER images were taken around 10:30 am in local time, we agree that the linear temperature profile cannot apply to over freezing glaciers in cold high mountains. We have addressed this issue with a reference in Discussion.

P10,L1-2: In addition, a non-linear temperature profile before the ice under the debris started to melt, which is common in the morning for debris in high-elevation continental glaciers, could produce additional errors.

➢ We have added information on grid cells with surface temperature < 0℃ to our product to notice pixels potentially having the issue of 0-degree assumption on the surface of the debris layer as well as the issue of non-linear temperature profile in freezing debris. The percentage of pixels with surface temperature < 0 degree in total glacier area varies from 14 % (New Zealand) to 89 % (Svalbard). We cannot judge whether the areas with surface temperature < 0 degree is covered by snow or clean ice, or cold debris layer, although most of these areas are in accumulation area where debris may not exist. We have added this issue in main text:

P9, L24-27: The percentage of such glacier area varies from 14 % (New Zealand) to 89 % (Svalbard) of total analysed glacier area and we cannot judge whether these areas are covered by freezing debris, although most of glacier area with low surface temperature often locate in accumulation area where debris may not exist.

I do want to express my support of this work as I believe mapping the thermal resistance (or debris thickness) of all glaciers would be a valuable contribution to the current state of knowledge; however, the severity of these underestimations is alarming and should be addressed.

➢ Thank you for your constructive comments. We addressed all potential limitation in our method in the main text.

**Response to Reviewer 1:**

(Referee comments #1)

General Comments

The study of Sasaki et al utilises multi-temporal satellite thermal imagery from the ASTER sensor and radiation data from the CERES project, combined with glacier outlines in the Randolph Glacier Inventory, to estimate: (1) the global distribution (excluding polar regions) of supraglacial debris cover at 90 m resolution; and (2) the distributions of 'thick' (ablation inhibiting) and 'thin' (ablation enhancing) debris cover over the same areas. These are ambitious aims, but such a global dataset would be of enormous value to earth science, and studies of glacier response to climate change in particular. Unfortunately, the study contains basic flaws in the methodology, relating to mistaken and untested assumptions about the surface energy balance of debris cover. These render the estimates of 'thick' and 'thin' debris cover distributions unreliable, while a much more rigorous evaluation of the critical threshold between clean ice and debris-covered ice would be needed to support the estimates of total debris covered area. These issues are explained in more detail below. Due to these significant problems, it would be misleading if the outputs from this study were incorporated into global glacier models to assess the effect of debris on glacier melt, as intended by the authors.

➢ Thank you for your detail comments with many suggestions to improve the manuscript. We have addressed all issues raised by reviewers. Regarding to the issue of simplification of surface energy balance, we have updated our result by taken latent and sensible heat flux into account and corrected underestimation due to the assumption of neglecting surface heat flux. We also added caution on the limitation of our method including assumption of linear temperature profile.

➢ To avoid misleading of the result of clean ice and debris-covered ice, we showed distribution of thermal resistance in different category without using the terms thin and thick debris. We also compared our result with previous studies more carefully and discussed potential reasons on differences.

➢ Our one-by-one reply to your detail comments are given below.

Specific Comments

Key Issue 1: Assumptions regarding the surface energy balance of supraglacial debris Section 2.2 of the paper introduces several important assumptions which undermine the so called 'energy balance' approach in this study. Only one of these assumptions (that the debris-ice interface is at 0 degrees C.) is assessed later in the paper, although its implications are not fully realised in the discussion.

The values of thermal resistance, R, which are the basis of the mapping methodology, are derived as an energy balance residual and in order to do achieve this, the energy balance needs to be as complete, consistent and as accurate as possible. The methodology fails to achieve this. The assumptions of a) instantaneous linear temperature gradient, and b) no heat storage in the debris layer are known to be incorrect from several studies (e.g. Nicholson and Benn, 2006; Brock et al., 2010, Reid and Brock, 2010; Rounce and McKinney, 2014). These processes are spatially and temporally variable in magnitude, depending on debris thickness, thermal properties and weather conditions before and at the time of satellite imaging and yet these important assumptions are not revisited in the paper. This undermines the attempt to estimate areas of debris cover with different melt impacts (i.e. 'thick' and 'thin' debris) made in the paper.

➢ The issue of the 0-degree assumption in debrise-ice interface is revisited in the same paragraph and we have added index of pixels < 0-degree in our product:
   P9, L23-14: Regarding the limitation of the 0°C assumption at the debris-ice interface, we added information on grid cells with a surface temperature < 0°C to our product.

➢ We agree that the assumption of linear temperature profile can be applied to debris with thickness up to 0.10 ~ 0.50 m. For example, Foster et al. (2012) showed relation between modeled debris thickness and surface temperature derived from ASTER and found that surface temperature is sensitive to debris thickness when debris is thinner than 0.50 m. Nicholson and Benn (2012) showed that observed temperature profile of debris layer reflects diurnal variation of surface temperature up to 0.50 m in depth. Rounce and McKinney et al. (2014) indicated that linear temperature profile can be applied in the upper 10 cm of debris layer in morning (10:15 local time). Schauwecker et al. (2015) concentrated several studies which investigated the temperature profile in debris layer, and we found that linear temperature profile can be applied until 20 cm from Fig. 3 of them. These researches indicated that our assumption of linear temperature profile can be applied to debris layer < 0.10 ~ 0.50 m.

➢ Despite the underestimation of thermal resistance of debris with thickness > 0.10 ~ 0.50 m, we can at least provide information of thermal resistance distribution up to 0.10 m and indication of areas with debris > 0.10 m. Since insulation effect of debris is dominant in the debris layer > 0.10 m, our thermal resistance product would be still valuable to apply studies such as global glacier models to include. We therefore would like to provide our thermal resistance product with clear notion of the underestimation of thermal resistance in thick debris > 0.1~0.5 m in Discussion.
   P9, L29-P10, L5: The assumption of a linear temperature profile in the debris layer is an additional source of uncertainty. Several studies indicated that a linear temperature profile can be applied to debris < 0.50 m, which is equivalent to thermal resistance > 1.43 $m^2KW^{-1}$ (*i.e.*, for ash, thermal conductivity = 0.35 $Wm^{-1}K^{-1}$) or > 0.5 $m^2KW^{-1}$ (others, thermal conductivity =

1.00 Wm$^{-1}$K$^{-1}$) (Foster et al., 2012; Nicholson and Benn, 2012). Rounce and McKinney (2014) indicated that the linear temperature profile can be applied to debris thinner than 0.10 m in glaciers in the Nepalese Himalayas (thermal conductivity = 0.96 Wm$^{-1}$K$^{-1}$). The estimated thermal resistance will be underestimated if the debris layer is thicker than these thresholds. Because insulation effects of debris are dominant with thicknesses > 0.10 m, and because we defined thick debris as thermal resistance > 0.25 m$^2$KW$^{-1}$ in the above analysis (Fig.8), which is smaller than the limitation of the linear temperature profile, the estimated thermal resistance above the threshold provides information on relatively thick debris, despite the underestimation of thermal resistance.

> We have added limitation of the assumption of no heat storage with its effect to underestimate thermal resistance in thick debris in Discussion.
> P10, L11-14: We assumed no heat storage in the debris layer, while Foster et al. (2012) suggested that more than half of the incoming energy is stored in the debris layer in the Mirage glacier in Italy. Because the heat storage of the debris layer changes depending on the thickness or type of materials comprising debris, it is difficult to set a global value of the heat conductivity to incorporate this effect. This limitation will cause underestimation of thermal resistance, particularly in thick debris.

The next assumption which is not evaluated is that the emissivity of debris = 1. Given rock forming minerals all have emissivity < 1 this assumption is clearly wrong, and even very small changes in emissivity of 0.01 have a significant impact on the calculated longwave radiation flux estimated from the Stefan-Bolzmann relationship (equation 3). There is no acknowledgement of this problem or attempt to evaluate its impact on debris cover estimations in the paper.

> We have changed the emissivity from 1.00 to 0.95 based on references (Nicholson and Benn, 2012; Rounce et al., 2015) and recalculated thermal resistance using the new emissivity. As a result, maximum thermal resistance in Baltoro Glacier increase is negligible. See figures below. We have acknowledged potential error range due to the assumption of single emissivity value for whole global in discussion.
> P10, L15-18: We assumed a single value of emissivity (0.95) for the entire globe. This may cause additional uncertainty. The emissivity of debris varies, depending on the type of debris. For example many studies used emissivity values of 0.94 or 0.95 for debris in Asia and Europe (e.g., Nicholson and Benn, 2012; Rounce et al., 2015), while 1.00 was assumed in the case of ash debris (Reid and Brock, 2010). These differences in emissivity are almost negligible in terms of the estimated thermal resistance.

[Figure]

Figure A: An example of thermal resistance in Baltoro with different emissivity.

Probably most critical is the assumption that turbulent fluxes = 0. The 4 references given here are selective, dated and do not provide support for this assumption. Recent, and more rigorous, field and modelling studies of the energy balance of supraglacial debris have been ignored (e.g. Nicholson and Benn, 2006; Brock et al., 2007, Brock et al., 2010, Reid et al., 2012; LeJenue et al., 2013; Collier et al., 2014; Fyffe et al., 2014; Rounce et al., 2015). These studies clearly demonstrate the opposite, i.e. that the sensible heat flux is a very large flux of energy away from the debris surface, particularly under the conditions when the imagery will have been captured, i.e. debris is being warmed strongly by insolation leading to a steep debris-air temperature gradient. Furthermore, the importance of including turbulent fluxes when mapping debris cover from thermal satellite imagery using an 'energy balance residual' approach has been demonstrated by several studies (Foster et al., 2012; Rounce and McKinnery, 2014;  Schauwecker et al., 2015). The assumption of zero turbulent fluxes creates an energy imbalance and a large underestimate of the thermal resistivity of debris covers (R; equation 4). This problem can be demonstrated by substituting the resulting R values (e.g. figures 3-5) into equation 1, using possible thermal conductivity (lambda) values of between 0.5 and 2, to estimate debris thickness. The resulting debris thickness is only in the range up to a maximum of 14 cm for lambda = 2, and 7 cm for a more realistic value of lambda = 1 (or 3.5 cm for lambda = 0.5). For example, on Baltoro glacier debris is estimated to be only a few cm over the majority of the glacier ablation zone (Figure 3) while it is known  to exceed 100 cm on the lower tongue. The other examples in figures 3 to 6 show similarly unrealistically low debris thickness values if realistic thermal conductivity values are used.

Crucially, errors due these assumptions will be spatially and temporally variable due to variable meteorological conditions (particularly wind speed and air temperature) and debris thickness and thermal properties. This renders not only the resulting R values physically meaningless and unusable, but also makes relative comparisons between different glaciers, and even different sites on the same glacier, impossible. Hence, the key aim of the paper, to produce a global map of the thermal resistance of debris for use in global glacier models is not achieved. The 'headline' claim that the global area of thick debris (ablation inhibiting) is two times the area of thin debris (ablation enhancing) is unsupported by the analysis.

➢ Regarding to the issue of simplification of surface energy balance, we have updated our method by taken latent ant sensible heat flux into account. As a result, for example, maximum value of thermal resistance in Baltoro glacier increases 44% (from 0.09 to 0.13). Although overall distribution of thermal resistance does not change in the new calculation, underestimation of thermal resistance has improved in many regions.

➢ The large underestimation of thick debris in Baltoro glacier is mainly due to the assumption of linear temperature profile in debris layer. As we discussed in the previous reply, we underestimate thermal conductivity of debris layer < 0.1~0.5 m when we use the assumption of linear temperature profile in debris layer. Since insulation effect of debris is dominant in the debris layer > 0.10 m, our thermal resistance product would be still valuable to apply studies such as global glacier models to include. We therefore would like to provide our thermal resistance product with clear notion of the underestimation of thermal resistance in thick debris > 0.1~0.5 m in Discussion.

P6, 22-24: We also underestimated the debris thermal resistance in the lower terminus, mainly due to the limitation in the linear temperature profile of the debris layer, which is discussed in Section 4.3.

P9, 29 – P10, L1: The assumption of a linear temperature profile in the debris layer is an additional source of uncertainty. Several studies indicated that a linear temperature profile can be applied to debris < 0.50 m, which is equivalent to thermal resistance > 1.43 $m^2KW^{-1}$ (*i.e.*, for ash, thermal conductivity = 0.35 $Wm^{-1}K^{-1}$) or > 0.5 $m^2KW^{-1}$ (others, thermal conductivity = 1.00 $Wm^{-1}K^{-1}$) (Foster et al., 2012; Nicholson and Benn, 2012). Rounce and McKinney (2014) indicated that the linear temperature profile can be applied to debris thinner than 0.10 m in glaciers in the Nepalese Himalayas (thermal conductivity = 0.96 $Wm^{-1}K^{-1}$). The estimated thermal resistance will be underestimated if the debris layer is thicker than these thresholds.

➢ Because our threshold of thick debris (> 0.25 $m^2KW^{-1}$) is smaller than the upper limit of the applicability of linear temperature profile (1.43 $m^2KW^{-1}$ for case of ash with thermal conductivity = 0.35 $Wm^{-1}K^{-1}$ or 0.5 $m^2KW^{-1}$ for others with thermal conductivity = 1.00 $Wm^{-1}K^{-1}$), the relative difference in thick debris distribution reflects reality despite the underestimation of thermal resistance in debris thicker than 0.10 - 0.50 m. On the other hand, our definition of thin debris is subjective and include dirty ice, patchy debris or thin debris with low thermal conductivity. We therefore showed distribution of thermal resistance in different category without using the terms thin and thick debris in abstract and results to avoid confusion.

Key Issue 2: Threshold thermal resistance between clean ice and debris-covered ice
This threshold R value is set to 0.01 m^2 K W^-1 (p. 5, line 15). Note that, given the problems discussed under Key Issue 1, there is large uncertainty in derived R values and this critical value will not provide a globally-consistent clean ice-debris cover threshold between different glaciers and regions. Given that this threshold value is critical to the estimates of total debris covered area, there is surprisingly little discussion of what this value means in terms of actual debris cover, and what the implications of uncertainty in the

critical threshold are. What is the effect of changing the R threshold to 0.02, or to 0.001, for example? The value of 0.01 sounds like a convenient number, rather than one that is based on sound scientific reasoning. The upper ablation zones of debris covered glaciers typically have large areas of dirty ice, patchy debris, and thin debris cover which grades into mostly continuous debris cover down glacier. It is not clear what the 0.01 threshold corresponds to in this transition. This is an important issue as thin and patchy debris is likely to lead to increased melt rates compared with clean ice, whereas continuous debris will normally lead to a reduction in ablation. Uncertainty in where this cut-off lies undermines the estimates of total debris covered area, and further weakens the attempt to calculate relative areas of 'thick' and 'thin' debris.

➢ We defined threshold between debris-covered and clean ice based on thermal conductivity given by references when we compared our result with other studies. We then set globally uniform threshold the thermal resistance as low (0.01–0.015 $m^2KW^{-1}$), intermediate (0.015–0.025 $m^2KW^{-1}$), or high (≥0.025 $m^2KW^{-1}$) to compare relative difference of thin, intermediate, and thick debris among different regions. The threshold of high does not affect the relative difference in thick debris, however, as you indicated, the threshold of low (0.01 $m^2KW^{-1}$) affects the percentage of thin debris and clean ice. Moreover, because we observed average value of thermal resistance in 90 m pixel, the pixel classified as clean ice includes dirty ice, patchy debris and thin debris.

➢ To avoid misleading of the result of clean ice and debris-covered ice, we have showed distribution of thermal resistance in different category without using the terms thin and thick debris in abstract and results:

P1, L16-20: We found that supraglacial debris with thermal resistance > 0.10 $m^2KW^{-1}$ covered 17.4% of the entire glacial area analyzed. The highest debris cover percentage occurred in New Zealand, and the lowest was in Iceland. The area of high thermal resistance (> 0.25 $m^2KW^{-1}$, which reflects a relatively thick debris layer, often suppressing glacier melt) was about two times that of debris with low thermal resistance (0.10~0.15 $m^2KW^{-1}$, which reflects relatively thin debris, often accelerating glacier melt), while 82.6% of the glacier area was relatively clean (< 0.01 $m^2KW^{-1}$).

➢ We have added that our definition of clean ice includes several conditions:

P2, L25-26: Here, we define debris-covered glacier as ice with high thermal resistance which includes pebbles, stones and rocks in glacier terminus, medial moraines, dirty ice, patchy debris and thin debris.

P8, L12-13: Note that we define debris cover as ice with high thermal resistance, which includes the effects of dirty ice, which is not defined as debris in previous observational studies.

➢ We have also addressed the issue of observed average thermal resistance in 90 m pixel and corresponding debris condition varies in reality.

P10, L33–P11, L5: We detected debris as high thermal resistance in 90 m pixel satellite images. Because thermal resistance is highly variable within a pixel in reality, the observed thermal resistance reflects several different conditions of the ice surface. For example, decreasing the debris area without changing thickness, and decreasing the thickness without changing the area of debris, may appear as similar changes in thermal resistance. Clean ice between thick debris may be detected as debris-covered ice when it is scaled up to 90 m pixels. Bare rock at the glacier boundary, melt ponds, shadows, ice cliffs or highly crevassed areas could also affect high thermal resistance.

The evaluation of debris covered areas in figures 3-6 shows some general agreement between the present study and earlier work, but also important discrepancies, particularly missing medial moraines.

➢ We have compared our result with previous research more carefully and included agreement and discrepancies with potential reasons more in detail in text.

P6, L18-24: Our results from the Baltoro Glacier (Karakoram) (Fig. 3a) were in close agreement with those of Mihalcea et al. (2008), who estimated thermal resistance by combining ASTER satellite images, meteorological data, and field measurements. However, our method could not resolve the medial moraine in the south branch of the upper area. This discrepancy may arise from our assumption of a linear temperature profile in the debris layer, while Mihalcea et al. (2008) assumed an exponential function for the surface temperature to derive thermal resistance. We also underestimated the debris thermal resistance in the lower terminus, mainly due to the limitation in the linear temperature profile of the debris layer, which is discussed in Section 4.3.

P7, L15-20: Visual comparison of the extent of debris cover showed similar distributions at all five sites, specifically in the debris distributed along the glacier tongue and flanks. Some narrow medial moraines are not resolved in our study due to the limited resolution of satellite images. For the Oberaletsch glacier, our estimated debris distribution is close to that in Paul et al. (2004). For the Haut Glacier d'Arolla, our estimated debris cover successfully reproduced the debris in the west and east flanks reported by Reid et al (2012), while the narrow medial moraine was not detected due to limited spatial resolution of the satellite images.

The explanation for the 'missing' debris cover at Koxkar Glacier (some 18 km^2 of debris and 60% of the ablation zone according to Juen et al., 2014) discussed in Section 4.3 is reasonable, but this does demonstrate the problem that, for high elevation continental

glaciers, overnight freezing of debris is common (particularly under clear sky conditions) which is a significant issue for the energy balance method of debris thickness estimation as it leads to non-linear temperature profiles in debris and significant changes in energy stored debris (both sensible and latent) as the debris warms in the morning (when satellite data are acquired). These processes are unaccounted for in the methodology.

- As we included sensible and latent heat in energy balance calculation, our estimation becomes better than the previous version.

- Since the ASTER images were taken around 10:30 am in local time, we agree that the linear temperature profile cannot apply to over freezing glaciers in cold high mountains. We have addressed this issue with a reference in Discussion.

  P10, L9-10: In addition, non-linear temperature profile before ice under debris started melt, which is common in morning of debris at high elevation continental glaciers, could produce additional errors.

- We have added information on grid cells with a surface temperature < 0-degree to our product to notice pixels potentially having the issue of 0-degree assumption on the surface of the debris layer as well as the issue of non-linear temperature profile in freezing debris. The percentage of such region (< 0 degree in surface temperature) varies from 14% (New Zealand) to 89% (Svalbard). Most of such areas may distribute in accumulation zone where debris may not exist. We clarified this in main text:

  P10, L9-10: In addition, non-linear temperature profile before ice under debris started melt, which is common in morning of debris at high elevation continental glaciers, could produce additional errors.

  P9, L23-28: Regarding the limitation of the $0°C$ assumption at the debris-ice interface, we added information on grid cells with a surface temperature $< 0°C$ to our product. The percentage of such glacier area varies from 14 % (New Zealand) to 89 % (Svalbard) of total analysed glacier area and we cannot judge whether these areas are covered by freezing debris, although most of glacier area with low surface temperature often locate in accumulation area where debris may not exist. For example, about 75% of grid cells with a surface temperature $< 0°C$ distributed at elevation higher than equilibrium-line altitude (5200 m, reported by Bocchiola et al., 2011) at the Baltoro glacier.

Technical corrections
Overall, the study is well presented and concisely written. Given the significant problems with this work discussed above, however, I have not included a list of technical corrections in this review.

**Response to Dr. Frank Paul:**

(Referee comments #2)

**General comments**

The study by Sasaki et al. applies a method to classify debris-covered glaciers as developed in earlier studies to a large part of the global glacier inventory as available from the RGI (Randolph Glacier Inventory). In contrast to numerous other studies, the aim here is not to classify the debris-covered parts itself, but to distinguish between clean ice and debris-covered ice within given glacier boundaries. The resulting dataset is going beyond a simple debris yes/no map and instead provides distributed thermal resistance of the glacier-covered area. This information is most welcome for a large number of applications looking at the energy balance for calculations of mass balance and future glacier development. I am also fine with the simplified approach suggested here, as achieving near-complete global coverage within a reasonable computational effort must have drawbacks somewhere. For these reasons I find the study timely and an important contribution.

On the other hand, I find some major shortcomings in this study that are partly based on problems of already published earlier studies, including some that are not cited here. In part and indirectly, the other two reviewers have mentioned the problem as well. The main issue is the never clearly defined and increasingly misleading use of the term debris-covered glacier. The term is used widely and independent of the spatial completeness of the debris coverage or its thickness. To my knowledge neither a glacier with a medial moraine nor with some more debris near the terminus should be called a debris-covered glacier. This requires that larger parts (tbd) of the ablation area have a near-complete (tbd) coverage with optically thick pebbles, stones or rock. Millimetre-sized particles (sand, silt, clay, BC, pollutants) that are often creating the rough microstructure of the ice do not fall into this category, as they never cover the ice completely.

➢ Since our main target is to develop thermal resistance distribution at global scale, we defined threshold between debris and clean ice based on thermal conductivity, and hence, the clean ice in our study includes medial moraines, debris on the terminus or dirty ice. We have added our definition of the term 'debris-covered ice' clearly in text. P2, L25-26: Here, we define a debris-covered glacier as ice with high thermal resistance, which includes pebbles, stones and rocks in the glacier terminus, medial moraines, dirty ice, patchy debris, and thin debris. P8, L12-13: Note that we define debris cover as ice with high thermal resistance, which includes the effects of dirty ice, which is not defined as debris in previous observational studies.

- In addition, to avoid misleading of the result of clean ice and debris-covered ice, we have showed distribution of thermal resistance in different category without using the terms thin and thick debris in abstract and results:

  P1, L16-20: We found that supraglacial debris with thermal resistance $> 0.10$ m$^2$KW$^{-1}$ covered 16.8% of the entire glacial area analyzed. The highest debris cover percentage occurred in New Zealand, and the lowest was in Iceland. The area of high thermal resistance ($> 0.25$ m$^2$KW$^{-1}$, which reflects a relatively thick debris layer, often suppressing glacier melt) was about two times that of debris with low thermal resistance ($0.10 \sim 0.15$ m$^2$KW$^{-1}$, which reflects relatively thin debris, often accelerating glacier melt), while 83.2% of the glacier area was relatively clean ($< 0.01$ m$^2$KW$^{-1}$).

- We have also added that our definition of clean ice includes several conditions:

  P2, L25-26: Here, we define debris-covered glacier as ice with high thermal resistance which includes pebbles, stones and rocks in glacier terminus, medial moraines, dirty ice, patchy debris and thin debris.

  P8, L12-13: Note that we define debris cover as ice with high thermal resistance, which includes the effects of dirty ice, which is not defined as debris in previous observational studies.

  P11, L3-5: Clean ice between thick debris may be detected as debris-covered ice when it is scaled up to 90 m pixels. Bare rock at the glacier boundary, melt ponds, shadows, ice cliffs or highly crevassed areas could also affect high thermal resistance.

A side issue of this problem is that the aggregation of often highly variable thermal information at the level of a 90 m thermal pixel is not addressed in most studies, e.g. by looking at least at the co-registered spectral information at 15 and 30 m resolution (speaking of ASTER). In consequence, an increasing amount of clean ice (towards higher elevations) within a 90 m pixel is often confused with apparently decreasing thickness of debris cover (although a glacier is never sorting particle sizes).

- We agree that we are observing average condition of inter 90m pixel. Since our result agree with our in-situ observation in several other glaciers, we believe that we provide meaningful information of debris distribution at large scale.

- We included several potential issues in observing average condition of 90m pixel in main text:

  P10, L33-P11, L5: We detected debris as high thermal resistance in 90 m pixel satellite images. Because thermal resistance is highly variable within a pixel in reality, the observed thermal resistance reflects several different conditions of the ice surface. For example, decreasing the debris area without changing thickness, and decreasing the thickness without changing the area of debris, may appear as similar changes in thermal resistance. Clean ice

between thick debris may be detected as debris-covered ice when it is scaled up to 90 m pixels. Bare rock at the glacier boundary, melt ponds, shadows, ice cliffs or highly crevassed areas could also affect high thermal resistance.

I exemplify the problem here for Hailuogou Glacier (HG), as it has been used for the methodological development that is forming the base of this study (Zhang et al. 2011). HG might be considered as a debris-covered glacier, but in fact the (optically thick) debris is only covering its lowermost part. For the largest part of the ablation area the heavily crevassed surface is *dirty* but not really debris covered. On a micro-scale, the dirty ice is not covering the ice completely and a variable amount of zero-degree bare ice is contributing to the thermal signal when aggregated at 90 m pixels. For HG this issue is strongly enhanced due to the highly crevassed surface that adds further zero-degree zones to the 90 m pixel. In fact, the 15 m ASTER VNIR image shown in Fig. 3a by Zhang et al. (2011) also clearly reveal (despite the resolution limits) that HG is not really debris covered but the ice is dirty at best (see also tourist photos in Google Earth). Accordingly, the thermal information leads to a very low thermal resistance and high melt rates (Fig. 6a/b). My key question here is: How could the thermal resistance derived from a dirty/crevassed glacier serve as a base for characterizing resistance for really debris-covered glaciers (let alone to determine debris thickness)?

➢ The aim of this study is to provide thermal resistance distribution at large scale. Our method cannot distinguish debris-covered ice and dirty ice as well as the effect of bare rocks at boundary of glacier outline or highly crevassed area. We included our definition of debris-covered glacier in an abstract, as well as these limitations in discussion:

P2, L25-26: Here, we define a debris-covered glacier as ice with high thermal resistance, which includes pebbles, stones and rocks in the glacier terminus, medial moraines, dirty ice, patchy debris, and thin debris.

P10, L33-P11, L5: We detected debris as high thermal resistance in 90 m pixel satellite images. Because thermal resistance is highly variable within a pixel in reality, the observed thermal resistance reflects several different conditions of the ice surface. For example, decreasing the debris area without changing thickness, and decreasing the thickness without changing the area of debris, may appear as similar changes in thermal resistance. Clean ice between thick debris may be detected as debris-covered ice when it is scaled up to 90 m pixels. Bare rock at the glacier boundary, melt ponds, shadows, ice cliffs or highly crevassed areas could also affect high thermal resistance.

When looking at Fig. 6 in Zhang et al. (2011) there is another severe issue in their analysis: A large portion (maybe not all) of the pixels indicating a high thermal resistance (the red-orange band) are actually located outside the glacier boundary and cover the lateral

moraine (and regions above) that are especially warm due to their south-easterly exposition and steep slope, i.e. the sun might hit these surfaces under a zero-degree incidence angle. There is definitely no ice underneath this "debris" that cools it. In consequence, the derived equations / regressions make no sense and could not be applied. This mistake highlights another problem with the (coarse) 90 m pixel of the thermal band. Not a single 90 m pixel with parts outside the glacier boundary should have been used to determine thermal information, as even a very small part of this warm rock impacts considerably on the mean value for the 90 m pixel (the same applies in the other direction when some bare ice is included). When the determination of debris thickness on a glacier is derived from rocks in the lateral moraine and an unconsidered bare ice part in the 90 m pixels, I do not wonder about funny results. At least a scientific base is missing.

➢ As we obtain glacier outline given from RGI, we may observe bare rock in glacier boundary as debris-covered ice (with high thermal resistance) as you indicated. We visually checked to find these areas at several glaciers and found that this issue affects estimated thermal resistance in very limited number of grid cells. However, it is difficult to remove these pixels manually for whole global. We therefore have put an index of 'boundary' to the thermal resistance product and data user can distinguish the effect of bare rock in glacier boundary with help of other index of surface temperature < 0-degree. We included the issues of glacier boundary and effect of bare rock in the boundary in main text:

P10, L33: Uncertainty in the glacier outline of RGI could also affect this.

P11, L4-5: Bare rock at the glacier boundary, melt ponds, shadows, ice cliffs or highly crevassed areas could also affect high thermal resistance.

I agree that this might not be the correct place to criticise the former study by Zhang et al. (2011) or note that the peer-review system fails from time to time. But when such studies are used as a base for other studies so that their misleading results are multiplied, there should be a possibility to stop it. I am aware that the above might have consequences for several other already published studies but I do not ask here to withdraw them. However, I would highly appreciate if all scientists working on the thermal identification of debris-covered glaciers (debris extent, thickness, reflectance or whatever) would do it more carefully in the future and consider all effects playing a role. This includes ice cliffs, melt ponds, albedo variations, shadow, crevasses, clean ice between thick debris (in regions of incomplete coverage) and the accurate distinction between dirty ice and really debris-covered ice. All these features can be present within the limits of a single 90 m ASTER pixel.

➢ We agree the limitation of 90 m ASTER pixel to observe various different condition in ice surface. We have included all these issues in main text:

P10, L33-P11,L5: We detected debris as high thermal resistance in 90 m pixel satellite images. Because thermal resistance is highly variable within a pixel in reality, the observed thermal resistance reflects several different conditions of the ice surface. For example, decreasing the debris area without changing thickness, and decreasing the thickness without changing the area of debris, may appear as similar changes in thermal resistance. Clean ice between thick debris may be detected as debris-covered ice when it is scaled up to 90 m pixels. Bare rock at the glacier boundary, melt ponds, shadows, ice cliffs or highly crevassed areas could also affect high thermal resistance.

➢ Despite the limitations, our product of thermal resistance show similar distribution in observed debris distribution in several regions. This supports that our global data set of thermal resistance would increase the understanding of debris area distribution at large scale where not enough direct observation is conducted.

➢ We think that the work of Zhang et al. (2011) is a good challenge to obtain thermal resistance at large scale from satellite. From their Fig 4, the derived thermal resistance corresponds well with observed debris thickness in many places. Despite the limitations in methodology, we believe that estimation of thermal resistance at large scale is still valuable to increase our understanding of ice surface at many places where direct observation is limited.

My other main objection is the rather poor validation performed here. I fully appreciate that several examples are given to illustrate the performance of the method (Figs. 3-6), including those where the method did not work.

➢ We compared our result more carefully with other studies and clarified limitation more in detail. For example, we discussed difference in methodology to detect medial moraine in Haut Glacier d'Arolla.
P6, L18-24: Our results from the Baltoro Glacier (Karakoram) (Fig. 3a) were in close agreement with those of Mihalcea et al. (2008), who estimated thermal resistance by combining ASTER satellite images, meteorological data, and field measurements. However, our method could not resolve the medial moraine in the south branch of the upper area. This discrepancy may arise from our assumption of a linear temperature profile in the debris layer, while Mihalcea et al. (2008) assumed an exponential function for the surface temperature to derive thermal resistance. We also underestimated the debris thermal resistance in the lower terminus, mainly due to the limitation in the linear temperature profile of the debris layer, which is discussed in Section 4.3.
P7, 16-20: Some narrow medial moraines are not resolved in our study due to the limited resolution of satellite images. For the Oberaletsch glacier, our estimated debris distribution is close to that in Paul et al. (2004). For the Haut Glacier d'Arolla, our estimated debris cover successfully reproduced the debris in the west and east flanks reported by Reid et al (2012),

while the narrow medial moraine was not detected due to limited spatial resolution of the satellite images.

But the pure visual comparison gives a very unreliable base for a proper assessment. For example Baltoro Glacier in Fig. 3a: When I compare the thermal resistance map with a map of the pattern of clean ice / debris cover (nicely arranged in parallel medial moraines), I do not see any similarities. Upwards of the confluence area (Concordia) everything is completely blue despite several thick medial moraines, whereas rather clean ice (or again regions outside the glacier?) has a high thermal resistance. In the upper right it seems the glacier mask includes a larger rock outcrop (please note that the region in the red circle on Fig. 5 is also a rock outcrop; the glacier in this region melted away). To me it seems as if a satellite scene with a high amount of seasonal snow has been used for the classification, largely underestimating the real extent of debris cover. I am pretty sure this problem is prevalent also in many other regions.

➢ The discrepancy in the upward of the confluence area in Baltoro is mainly due to the threshold of color bar. After we changed the color bar, our estimated thermal resistance in Baltoro glacier corresponded well with debris information by Mihalcea et al. (2008) and Veetil et al. (2012) (see figure below). We have discussed that missing narrow medial moraines in upward of the confluence area that Mihalcea et al. (2008) showed is due to the limitation in horizontal resolution and difference in methodology. This discrepancy may come from our assumption of linear temperature profile in debris layer. Mihalcea et al. (2008) assumed exponential function of surface temperature to derive thermal resistance and explained that they overestimated thermal resistance in particular thin debris (Table.1 in Mihalcea et al. 2008).

P6, L18-24: Our results from the Baltoro Glacier (Karakoram) (Fig. 3a) were in close agreement with those of Mihalcea et al. (2008), who estimated thermal resistance by combining ASTER satellite images, meteorological data, and field measurements. However, our method could not resolve the medial moraine in the south branch of the upper area. This discrepancy may arise from our assumption of a linear temperature profile in the debris layer, while Mihalcea et al. (2008) assumed an exponential function for the surface temperature to derive thermal resistance. We also underestimated the debris thermal resistance in the lower terminus, mainly due to the limitation in the linear temperature profile of the debris layer, which is discussed in Section 4.3.

[Figure]

Fig B. Thermal resistance in Baltoro glacier in our study with modified color bar.

➤ Thank you for your advice on the condition of red circle in Fig.5. We have checked the latest image of Google Map taken in 2017 and found that the red circle in Fig. 5 in Haut Glacier d'Arolla seems bed rock. Landsat image showed dark color as well, as you indicated. We agree that this glacier is melting rapidly and some parts of ice disappeared. As we used glacier outline from RGI, we think that the usage of the latest glacier inventory is indispensable to avoid discrepancy between observation and outline of glacier inventory. We noticed this issue in main text.

P7, L25-30: This discrepancy was due to the large debris-covered area of the southeastern tributary of the glacier (marked as a red circle in Fig. 5a and b), where Reid et al. (2012) indicated ice without debris. The Landsat image taken in 2005 showed a dark area distributed in the southern tributary (Fig. 5 b), suggesting that there was debris on the glacier at that moment. The latest (2017) image in Google Maps (not shown) suggested there was no glacier and bare rock was observed in this area. Since we use the RGI outline to define ice, the estimated thermal resistance taken from ASTER images in 2009-2013 may include bare rock where glaciers were rapidly retreating.

➤ In Baltoro and Haut Glacier d'Arolla, the effect of snow cover is negligible because we could obtain enough number of satellite images without snow and cloud cover in these areas. The discrepancy in upper Baltoro is mainly due to the threshold of color bar and comparison with study which overestimated thickness in thin debris. However, we have also noticed that the effect of snow affects underestimation of thermal resistance in heavily snow region. As we used multiple images to obtain the highest thermal resistance to eliminate the effect of snow, inclusion of a greater number of satellite images could improve these errors in future.

Which brings me to a final major point, the selection of satellite scenes. I doubt that (a) the method applied here to select the most suitable scenes has always found the scenes with a minimum of snow cover and (b) I think for many regions suitable scenes simply do not exist (e.g. Fig. 6b). The selection of the scene with the largest amount of debris (P8, L21) is certainly a good idea but it does not imply that all debris is exposed. I think the uncertainty in this regard is not realistically estimated. I certainly miss a pixel-by-pixel comparison (omission and commission errors) based on a couple of manually created debris extents. This can be easily achieved by subtracting a clean ice classification (using a simple red/SWIR band ratio) from the RGI glacier extents (please use RGI 5.0!) converted to a grid. Such a clean ice mask would also help to determine the amount of clean ice within each 90 m thermal pixel and correct the details of the thermal resistance classification accordingly.

- Same as the case of Baltoro glacier and Haut Glacier d'Arolla, effect of snow on the estimated thermal resistance seems small in the case of Fig.6b due to the availability of good satellite images. As you indicated, Fig. 6b derived from Landsat is affected by snow but we did not use this image for our estimation. Figure C below is a satellite images we used to derive thermal resistance in our study. You can see that the effect of snow is minimum when we overlaid multiple ASTER images.

- Thank you for your advice for using clean ice mask from other data source. We included that our method have potential to apply other satellite images to increase the number of images to eliminate snow or cloud effect.

  P9, L21-23: We found that such regions accounted for less than 0.01% of the total analyzed area, and the inclusion of a greater number of satellite images, including other satellites (e.g., Landsat), could improve these errors in future.

- We have tried to use the latest RGI 5.0 in our revision, but we could not use "extraction" process in ArcGIS software may be due to the missing information of shapefiles. The files released by RGI has changed from RGI4.0 to RCI5.0 (only shape files are distributed in RGI5.0 while RGI4.0 distributed supplementary files with shape file). We have reported this issue to RGI data group in NSIDC and got a short reply that they are surveying the issue. We will update our product with RGI5.0 when it is available.

[Figure]

Figure C. ASTER images and derived thermal resistance in Caucasus region for Figure 6

Overall, I think the study was worth a try as it ultimately creates a dataset of high interest and demand. Given that the methodology is further improved, better validated and creating more realistic results, I am pretty sure that the study can be published. I would thus like to encourage the authors to check if they can improve their study along the suggested lines and resubmit it at a later stage. In the following I add some further points requiring consideration in the case the authors intend to resubmit the study.

➢ Thank you very much for your kind and constructive comments. We have addressed all these major concerns in the revision.

**Specific comments**

For future submissions, please use a continuous line numbering scheme and apply it to all lines rather than each 5th. This facilitates the work of reviewers greatly as they do not have to waste time with counting lines and page numbers.

As we used default file format given by a journal, the line numbering was given every 5 lines. We have changed the setting and put continuous numbering in the revision.

P1

L17-19: As a more general comment: Please note that the use of the simplified terrminology 'thin and thick debris' is misleading as it does not consider grain size and degree of coverage. The 'thin debris' that is enhancing surface melt is often just dirty ice (sand, clay, silt) with lots of clean ice in-between (i.e. incomplete spatial coverage).

We did not use the term "thin and thick debris" and used thermal resistance with thresholds to present our findings in abstract.

Moreover, for thin plates of rock (<2 cm) the albedo of the material is increasingly important. Please also note that surface melt is not the only factor contributing to volume/mass loss of glaciers and thus the amount of melt water. Several studies using DEM differencing have shown that specific mass loss of heavily debris-covered glaciers is often as high as for clean glaciers. Ice cliffs, melt ponds and so far completely unconsidered en-glacial melt (internal collapse of conduits) might play a major role for this. In short, the impact of debris cover on the amount of melt water from glaciers is not fully clear.

We agree that the impact of debris cover on the amount of melt water from the glacier is not clear comparing to other factors at this moment. Hence, we thought it is important to develop distribution of thermal resistance at large scale to investigate the effect of surface glacier condition on surface ice melt. Although we stated that debris effect is important, it does not exclude the importance of other factors on glacier melting. Because this paper is focusing on debris on glaciers only, we think that inclusion of other effects in Introduction is

confusing.

L24: Please just write 'glaciers'. Those contributing to sea level are certainly not 'small' but very large.

Revised as suggested.

L28: How does melt water from glaciers cause rock slides?

It is landslide, sorry.

P2

L4: Please note that glacier retreat (change in length) is something different than volume loss caused by the melting rate. Glaciers can lose mass without a change in length and advance without a change in mass. Global glacier models have a good hand on melt rates but difficulties with geometric changes, as this requires precise knowledge of the glacier bed (ice thickness distribution).

We used volume loss rather than retreat.

L7: As mentioned above, I would not call the material that is enhancing melt 'thin debris' but dirty ice, as grain size must be small, albedo low and its distribution disperse.

We would like to refer previous study as it is, since this general description of debris effect do not exclude the importance of other materials. We clearly defined our usage of debris cover as ice with high thermal resistance which include all potential condition of forcing higher thermal resistance.

P2, L25-26: Here, we define debris-covered glacier as ice with high thermal resistance which includes pebbles, stones and rocks in glacier terminus, medial moraines, dirty ice, patchy debris and thin debris.

P8, L12-13: Note that we define debris cover as ice with high thermal resistance, which includes the effects of dirty ice, which is not defined as debris in previous observational studies.

P11, L3-5: Clean ice between thick debris may be detected as debris-covered ice when it is scaled up to 90 m pixels. Bare rock at the glacier boundary, melt ponds, shadows, ice cliffs or highly crevassed areas could also affect high thermal resistance.

L13: Please use glacier instead of glacial when reference is made to contemporary glaciers (or here The melting of ice beneath debris …).

Revised as suggested.

L31: Please see my comments above on the study by Zhang et al. (2011). I think it contains major systematic and methodological errors and cannot be used as a reference.

We deleted 'accurately' because of several issues in their paper as you indicated (e.g., effect of crevasse). We, however, think that their work is a good challenge to obtain thermal resistance at large scale from satellite. From their Fig 4, the derived thermal resistance corresponds well with observed debris thickness in many places. Hence, we would like to remain their reference here.

P3

L4: As the thermal resistance map is obviously not able to distinguish clean ice from dirty ice and debris-covered ice, what about starting with a simple debris cover yes-no map?

We would like to provide our thermal distribution map to public, since many researchers are interested in the product. As you suggested, we clarified all potential limitation including distinguishing clean ice or underestimation of thermal resistance in thick debris. Other researchers can utilize the information of thermal resistance in their target region taking local ice condition into account. We think provision of thermal resistance values rather than debris cover yes/no map could contribute further understanding of debris-covered glaciers. Moreover, as indicated by all reviewers, definition of debris cover from thermal resistance value is difficult, because we do not have global distribution of thermal conductivity.

L9: Please use RGI 5.0.

As explained above, we could not use RGI 5.0 due to the file errors they distributed. We used RGI4.0 but will update our product when the issue in RGI5.0 is solved.

P5

L14/15: The threshold values of thermal resistance for debris yes/no and thick/thin should be provided in the methods section.

We moved these sentences at the end of method.

P6

L4: Can 'close agreement' be described in more detail? I did not find a very good agreement between the thermal resistance map and the distribution of clean ice and debris for Baltoro glacier (see General comments).

As we answered in General comments, we discussed agreement and disagreement with potential reasons in text.

P7

L1: This is not a debris-covered tributary but bare rock. That this region has been mapped

as debris-covered ice might be a consequence of the calibration over bare rock rather than debris-covered ice.

As we answered in General comments, we discussed that the place is bare rock in main text.

L11: Please note, debris cover should only appear on the surface below the ELA (or as a proxy: mean elevation) due to emergent flow. Apart from the higher amount of snow cover in the study by Lambrecht et al., I assume that your method has simply mapped dirty ice as being debris covered. Dirty ice is rather common in this region below steep (and ice free) rock walls.

We have added our definition of debris cover and dirty ice is other potential reason of discrepancy.

P8, L12-13: Note that we define debris cover as ice with high thermal resistance, which includes the effects of dirty ice, which is not defined as debris in previous observational studies.

In addition, we have added information of grid cells with surface temperature < 0-degree in our thermal resistance product. The additional information will help data uses to distinguish areas with low surface temperature where mostly located above the ELA line.

L22: by reducing albedo? I do not think that albedo reduction is the major process here. The key point is that the material on the surface can get warmer than zero degrees and when the material is thin enough, conductivity can get the base of the material also above zero degrees so that it can melt into the ice.

Schauwecker et al. (2015) and other many previous researches indicated that when ice is covered with thin debris, melting is enhanced due to the increased absorption of solar radiation due to the lower albedo and heat conductivity of the material comparing to ice. As you indicated, if the material of debris is the same, the main difference between thin and thick debris is difference in conductivity of surface energy to ice. We have clarified this in text.

P8, L16-19: The effect of debris on glacial melting depend on the thickness of the debris layer. If a debris layer is thin enough, it enhances melting since the base of debris gets warmer due to increased absorption of solar radiation with lower albedo, while a thick debris layer inhibits melt because energy conduction to the ice is reduced (e.g., Østrem et al., 1959; Driedger et al., 1981; Mattson et al., 1993).

L23ff: This sounds like methods rather than discussion.

We have moved the sentence to methods.

P9

L5: Please note that also the RGI outlines are not perfect and please avoid including any thermal pixels that reach beyond given outlines. Only 100% inside pixels should be used for the calculation.

We have included uncertainty in RGI outlines. We visually checked thermal resistance values in RGI outlines and found that there is some large values maybe due to the effect of bare rocks. However, number of these pixels are very limited and do not affect our result. We therefore added index glacier outline in our published thermal resistance data.

P10, L33: Uncertainty in the glacier outline of RGI could also affect this.

P17, Fig. 3: I suggest adding also here optical images for comparison (those with minimum snow amount).

We have included optimal images of Fig. 3 as suggested.